# The Role of Primary Cilia in the Crosstalk between the Ubiquitin–Proteasome System and Autophagy

**DOI:** 10.3390/cells8030241

**Published:** 2019-03-14

**Authors:** Antonia Wiegering, Ulrich Rüther, Christoph Gerhardt

**Affiliations:** Institute for Animal Developmental and Molecular Biology, Heinrich Heine University, 40225 Düsseldorf, Germany; antonia.wiegering@hhu.de (A.W.); ruether@hhu.de (U.R.)

**Keywords:** protein aggregation, neurodegenerative diseases, OFD1, BBS4, RPGRIP1L, hedgehog, mTOR, IFT, GLI

## Abstract

Protein degradation is a pivotal process for eukaryotic development and homeostasis. The majority of proteins are degraded by the ubiquitin–proteasome system and by autophagy. Recent studies describe a crosstalk between these two main eukaryotic degradation systems which allows for establishing a kind of safety mechanism. If one of these degradation systems is hampered, the other compensates for this defect. The mechanism behind this crosstalk is poorly understood. Novel studies suggest that primary cilia, little cellular protrusions, are involved in the regulation of the crosstalk between the two degradation systems. In this review article, we summarise the current knowledge about the association between cilia, the ubiquitin–proteasome system and autophagy.

## 1. Introduction

Protein aggregates are huge protein accumulations that develop as a consequence of misfolded proteins. The occurrence of protein aggregates is associated with the development of neurodegenerative diseases, such as Huntington’s disease, prion disorders, Alzheimer’s disease and Parkinson’s disease [1,2,3], demonstrating that the degradation of incorrectly folded proteins is of eminent importance for human health. In addition to the destruction of useless and dangerous proteins (protein quality control), protein degradation is an important process to regulate the cell cycle, to govern transcription and also to control intra- and intercellular signal transduction [4,5,6]. Two main protein degradation systems exist in vertebrates—the ubiquitin–proteasome system (UPS) and macroautophagy (hereafter referred to as autophagy) [7]. Their function is not only essential for vertebrate homeostasis but also for vertebrate development [4,8,9,10,11,12,13,14,15,16,17,18,19,20,21,22,23,24,25,26,27,28,29]. Importantly, the UPS and autophagy are, at least partially, redundant. If one degradation system is downregulated, the other gets upregulated to prevent cell-damaging protein overload or the formation of protein aggregates, as well as to ensure the maintenance of pivotal intra- and intercellular signalling [4,7,30]. However, the proteasome-to-autophagy direction of regulation is far better documented than the autophagy-to-proteasome direction [31]. Evidence for the autophagy-to-proteasome direction is mainly provided by investigations in cancer cells and in cultured neonatal rat ventricular myocytes [32,33], while numerous studies reported findings that support the existence of the proteasome-to-autophagy direction [34,35,36,37,38,39,40,41,42,43,44,45,46,47]. In any case, a kind of crosstalk takes place between the UPS and autophagy. This article focuses on the role of primary cilia in this crosstalk. In the following sections, we will shortly introduce the UPS, autophagy and primary cilia. Afterwards, we will discuss a potential role for the UPS and autophagy in cilia-associated diseases and mechanisms underlying the UPS–autophagy crosstalk with particular regard to primary cilia.

## 2. The Ubiquitin–Proteasome System and Autophagy

The vast majority of the proteins (~80–90%) within the vertebrate cell are degraded by the UPS [7]. Apart from the degradation of proteins, the UPS is able to implement the proteolytic processing of particular proteins [48,49]. During this process, one or more peptide bonds of the target protein are hydrolysed. Both protein degradation and protein processing, carried out by the UPS, start with the ubiquitination of target proteins. Ubiquitin conjugation is performed by a cooperative action of ubiquitin-activating enzymes (E1), ubiquitin-conjugating enzymes (E2) and ubiquitin ligases (E3). In simplified terms, ubiquitin is activated by E1 enzymes when ATP is present and, thereafter, is transferred to E2 enzymes. Two different types of E3 ligases exist: homologous to the E6-AP carboxyl terminus (HECT) domain E3 ligases and really interesting new gene (RING) finger domain E3 ligases. The E2 enzymes pass ubiquitin onto the HECT domain E3 ligases which transfer it to the target protein [50,51]. In contrast to the HECT domain E3 ligases, the E2 enzymes do not convey ubiquitin to the RING domain E3 ligases but directly to the proteasomal substrates. The RING domain E3 ligases act as a kind of bridge between the ubiquitin-bound E2 enzymes and the substrates, thereby increasing the activity of the E2 enzymes [52]. In the context of ubiquitination, three different models exist that explain the formation of a ubiquitin chain (polyubiquitination) bound at proteasomal substrates. The first model describes the formation of the chain in a step-by-step process in which ubiquitin monomers are added sequentially to the substrate. The second model states that the ubiquitin chain might be pre-assembled on an E2 enzyme and then transferred to the substrate in a single process. The third model represents a combination of the first two models [53,54,55]. Finally, polyubiquitinated proteins are degraded or processed by the catalytic component of the UPS, the 26S proteasome. The proteasome represents a large multi-protein complex of about 1700 kDa which comprises two different kinds of subunits—the 19S subunit and the 20S subunit (Figure 1A) [56,57]. The ubiquitin chain of target proteins is recognised and bound by the 19S regulatory complex and, subsequently, the target proteins are unfolded [58]. Hereafter, these proteins get degraded or processed by the 20S subunit which harbours different protease activities (caspase-like activity, chymotrypsin-like activity, trypsin-like activity) [59]. Proteasomes were detected in the cytosol, cell nucleus, microsomes, centrosomes and at the base of primary cilia [60,61,62,63]. Since their action is of great importance for the proper transduction of numerous signalling pathways [64,65], an altered proteasomal activity provokes defects in the regular procedure of cellular signalling and associated cellular processes [66], reflecting the eminent role of the UPS in the development and function of many vertebrate organs and structures [20,21,22].

In contrast to the UPS, autophagy is able to degrade proteins but cannot process them. However, autophagy degrades intracellular pathogens, long-lived proteins, very large protein complexes and even entire cell organelles [7,67,68,69,70,71,72]. In this way, autophagy plays an important role in vertebrate development [9,10,17,18,19], participating in the development of the brain [23], eyes [24,25,26], lung [27], heart [28,29] and liver [28]. Autophagy starts with the formation of phagophores, which then elongate to develop autophagosomes (Figure 1B) [73]. Elongation of the phagophore membrane and formation of the autophagosome is dependent on the recruitment of two ubiquitin-like (Ubl) conjugation systems. To build up the first system, autophagy-related protein 12 (ATG12) gets bound to ATG5 by the action of the E1-like enzyme ATG7 and the E2-like enzyme ATG10. Afterwards, ATG12–ATG5 becomes associated with ATG16L and forms a large complex referred to as the ATG16L complex. This complex is located at the phagophore, thereby defining the site of conjugation of the second Ubl system. The second Ubl conjugation system starts with the cleavage of microtubule-associated protein 1B light chain 3 (LC3) by ATG4 to generate cytoplasmic LC3-I. With the support of ATG7 and another E2-like enzyme called ATG3, LC3-I becomes lipidated with phosphatidylethanolamine (PE) to generate membrane-tethered LC3-II [74]. Autophagosomes represent double-membraned vesicles which enclose the targets destined to be degraded. LC3-II is integrated into both the inner and outer membrane of the autophagosome where it functions in autophagy substrate selection [75,76,77]. To degrade their enclosed substrates, the autophagosomes fuse with lysosomes [73]. The formed autolysosomes release hydrolases that implement the degradation of the autophagy substrates (Figure 1B). Autophagy is regulated by mammalian target of rapamycin (also known as mechanistic target of rapamycin) complex 1 (mTORC1) which blocks the initiation of autophagy. mTOR inactivates the ULK complex by phosphorylation [78,79]. The ULK complex consists of unc-51-like kinase 1/2 (ULK1/2), focal adhesion kinase family-interacting protein of 200 kDa (FIP200) and ATG13. When mTOR is inactivated, the ULK complex is allowed to translocate to phagophores and ULK1 directly phosphorylates Beclin-1, thereby activating the pro-autophagy class III phosphoinositide 3-kinase (PI(3)K) VPS34 complex and promoting autophagy induction and maturation [80].

As with the UPS, autophagy is also able to recognise a small proportion of its target proteins by their polyubiquitination (selective autophagy). Ubiquitin has different lysine residues which participate in the generation of polyubiquitin chains [81,82]. While the UPS preferably degrades or processes proteins that have a K48-linked polyubiquitin chain, autophagy degrades proteins with K63-linked chains [83,84].

## 3. The Primary Cilium

Primary cilia are tiny cytoplasmic protrusions (1–15 μm long) and basically contain three different compartments—the axoneme, the basal body (BB) and the transition zone (TZ) (Figure 1C). The axoneme represents the microtubule-based scaffold of the cilium and grows out of the BB, which is the modified mother centriole. The axoneme consists of nine doublet microtubules that are organised in a ring-like fashion. It stabilises the cilium and is essential for intraflagellar transport (IFT). IFT carries proteins from the base to the tip of the cilium (anterograde IFT), and then back to the base (retrograde IFT). The implementation of IFT requires the presence of motor proteins (kinesin-2 for anterograde IFT and dynein-2 for retrograde IFT) and of so-called IFT proteins. IFT proteins belonging to the IFT-B complex (e.g., IFT88) drive anterograde IFT, while IFT proteins of the IFT-A complex (e.g., IFT140) are necessary for retrograde transport [85]. The cargo that is destined for transport within the cilium is bound to IFT proteins which, in turn, are attached to motor proteins that move along the microtubules of the axoneme. The TZ, a short region of 0.5 μm, is located at the proximal end of the axoneme. It is characterised by the presence of so called Y-links, structures which appear as Y-shaped densities by transmission electron microscopy [86,87]. The TZ functions as a ciliary gatekeeper, controlling the entry and exit of proteins into and out of the cilium [87,88,89,90,91,92,93,94,95]. Several of the proteins that traverse the TZ and that are translocated through the cilium are receptors and mediators of signalling cascades such as the hedgehog (HH) pathway, the platelet-derived growth factor receptor α (PDGFRα) pathway and the transforming growth factor β (TGFβ) pathway [96,97,98,99,100,101,102,103]. HH signalling is one of the best studied cilia-mediated signalling pathways. In the absence of HH ligand, the activation of smoothened (SMO) is inhibited by the HH receptor patched (PTC), which is located in the ciliary membrane [101]. In this case, the HH mediator proteins glioblastoma 2 (GLI2) and glioblastoma 3 (GLI3) are proteolytically processed into GLI2-R and GLI3-R, two transcriptional repressors of HH target gene expression (Figure 2A). In the presence of HH, the HH ligand binds to PTC and the HH/PTC complex is translocated out of the cilium. Subsequently, SMO is activated, enters the cilium and induces the generation of GLI2-A and GLI3-A, two transcriptional activators of HH target gene expression [98,104,105] (Figure 2A). PDGFRα signalling starts with the binding of the ligand platelet-derived growth factor AA (PDGF-AA) to its ciliary membrane-bound receptor PDGFRα, which subsequently dimerises and undergoes phosphorylation [100]. As a consequence, the transduction of the mitogen-activated protein kinases 1/2 (MEK1/2)–extracellular signal-regulated kinases 1/2 (ERK1/2) and protein kinase B (AKT/PKB) pathways is initiated (Figure 2B). The importance of primary cilia in mediating PDGFRα signalling is reflected by the fact that the loss of cilia completely blocks PDGFRα signalling [100]. In the case of TGFβ signalling, both receptors of the pathway, TGFβ-RI and TGFβ-RII, localise to cilia [102]. By ligand binding, TGFβ-RI and TGFβ-RII form a heterotetrameric receptor complex which, in turn, activates the mediator proteins Sma- and Mad-related Protein 2 (SMAD2) and Sma- and Mad-related protein 3 (SMAD3) at the base of primary cilia (Figure 2B). Then, SMAD2 and SMAD3 join with Sma- and Mad-related protein 4 (SMAD4) at the ciliary base and the SMAD2–3–4 complex leaves the cilium in order to enter the nucleus and to induce TGFβ target gene expression [102]. Cilia-mediated signalling cascades are essential for the regulation of cellular processes during the entire development of an organism [103,106,107,108,109]. Consequently, many severe human diseases are associated with dysfunctional primary cilia and their number is permanently increasing [110]. The diseases caused by ciliary dysfunctions are commonly referred to as ciliopathies. Ciliopathies comprise many life-threatening diseases such as polycystic kidney disease, Meckel–Gruber syndrome, Joubert syndrome, Bardet–Biedl syndrome, Leber congenital amaurosis, Senior–Løken syndrome, orofaciodigital syndrome type 1, Alström syndrome, Jeune asphyxiating thoracic dystrophy, Ellis–van Creveld syndrome and Sensenbrenner syndrome [111,112].

## 4. Do the UPS and Autophagy Play a Role in the Development of Ciliopathies?

Current treatment of ciliopathies is limited to symptomatic therapies, as curative medication against ciliopathies is not yet available [113]. For this reason, cilia research is focused on the investigation of molecular mechanisms underlying ciliopathies as well as on developing curative therapies against these severe diseases [66,114,115,116,117,118,119,120,121]. There are promising approaches to tackle ciliopathies ranging from gene therapy to the use of small molecules, but none have yet successfully gone through clinical trials [122,123,124]. According to several studies, reduced activity of the UPS and/or of autophagy might be involved in the development of ciliopathies. For instance, it was reported that the ciliopathy phenotype of Bardet-Biedl syndrome (*bbs*) and Oral-Facial-Digital Syndrome 1 (*ofd1*) morphant zebrafish embryos is ameliorated by injecting human proteasomal subunit component (RPN10, RPN13, or RPT6) mRNA or by injecting the proteasome activators sulforaphane (SFN) and mevalonolactone (alias mevalonic acid lactone, mevalonate, and (±)-β-hydroxy-β-methyl-δ-valerolactone and abbreviated MVA), respectively [66]. Regarding autophagy, many data were collected in the context of polycystic kidney disease. In zebrafish embryos, it was demonstrated that a decreased autophagic activity causes polycystic kidney disease and that a specific inducer Beclin-1 peptide and the autophagy activators rapamycin (RAP) as well as carbamazepine (CBZ) and minoxidil ameliorates cyst formation and restores kidney function [125,126]. While the macrolide RAP activates autophagy by inhibiting mTORC1 signalling [127,128,129] which is known to block autophagy [130,131,132,133], carbamazepine (CBZ) and minoxidil exert their autophagy-activating function independently of mTOR [134,135]. Interestingly, in utero application of RAP also markedly attenuated cyst formation in mouse embryos suffering from polycystic kidney disease [136]. Moreover, treatment of adult mice suffering from polycystic kidney disease with RAP reduces renal cystogenesis [137,138]. Furthermore, the treatment of rats displaying polycystic kidney disease with RAP as well as with the mTOR inhibitor and autophagy activator PP242 blocks renal enlargement and cystogenesis [139,140,141,142]. In polycystic kidney patients, the application of RAP slows kidney growth and prevents the worsening of renal function [137,143,144,145]. In mice, it was shown that a proper dose of RAP is essential for its positive effect on polycystic kidney disease [146,147]. Based on these studies, it is conceivable that a decreased proteasomal and/or autophagic activity might be involved in the development of ciliopathies.

## 5. Which Role Does the Primary Cilium Play in the Crosstalk between the UPS and Autophagy?

For many years, crosstalk between the UPS and autophagy was negated since the general view was that the two main degradation systems have different substrate preferences [148,149]. In the last 10 years, interplay between the UPS and autophagy was the subject of intense research. The key finding was that the UPS and autophagy are at least partially redundant, a result that is based on the fact that both degradation systems partially share the same substrates, and that if one degradation system is downregulated, the other gets upregulated. It was shown that numerous proteins participate in this crosstalk [150]. Due to existing excellent review articles about the crosstalk between these degradation systems [31,150,151,152], we will concentrate on those interplay mechanisms in which the primary cilium is obviously involved.

It is unquestionable that the primary cilium takes part in the crosstalk between the UPS and autophagy, but it is difficult to define its exact role since different relationships between ciliary proteins and both degradation systems have been elucidated (Figure 3). BBS4 and OFD1 are good examples for explaining this difficulty. Both proteins positively regulate proteasomal activity since the loss of BBS4 (in the kidney, liver, brain and retina) and OFD1 (in mouse embryonic stem cells) results in a reduced proteasomal activity, respectively [66]. Moreover, both proteins are influenced by autophagy. In mouse embryonic fibroblasts (MEFs), BBS4 is recruited to primary cilia by autophagy, most likely via an indirect mechanism, and OFD1 is an autophagic substrate which is degraded at the ciliary base [153]. Considering that ciliary BBS4 positively regulates proteasomal activity, autophagy would promote proteasomal activity by the ciliary recruitment of BBS4. By contrast, autophagy seems to inhibit proteasomal activity via degrading OFD1. Furthermore, the autophagy-dependent degradation of OFD1 at the ciliary base promotes ciliogenesis and ciliary elongation [153,154]. Consequently, autophagy ensures the formation of cilia. In turn, ciliary presence is essential for the proteolytic processing of GLI3 in mice [99]. Investigations in MEFs revealed that GLI3 processing is implemented by the cilia-regulated proteasome, a kind of proteasome that localises to the ciliary base and is controlled differently from all other proteasomes within the cell [63]. The absence of retinitis pigmentosa GTPase regulator-interacting protein 1-like (RPGRIP1L) leads to a reduced catalytic activity of the cilia-regulated proteasome, while the activity of all other proteasomes within the cell is not affected [63]. The proteolytic processing of GLI3 gives rise to GLI3-R, which inhibits HH signalling [48]. Cilia-mediated HH signalling is able to activate autophagy in MEFs and mouse kidney epithelial cells [155]. Thus, the cilia-regulated proteasome negatively regulates autophagy. As mentioned before, the catalytic activity of the cilia-regulated proteasome is positively governed by RPGRIP1L in MEFs, suggesting that RPGRIP1L negatively regulates autophagy via its positive regulation of the cilia-regulated proteasome. However, RPGRIP1L positively regulates both the activity of the cilia-regulated proteasome and autophagy in MEFs. Additionally, it was demonstrated that RPGRIP1L controls both activities independently of each other in MEFs [156]. Remarkably, RPGRIP1L negatively controls the proteasome-based degradation of dishevelled (DSH) in Madin–Darby canine kidney (MDCK) cells [157], suggesting that RPGRIP1L governs proteasomal activity in a cell type-specific manner. Moreover, RPGRIP1L controls the assembly of the TZ by ensuring the proper amount of other proteins at the ciliary base in MEFs, mouse embryonic kidneys and mouse embryonic limbs [158]. One of these proteins is BBS4. In MEFs, autophagy recruits BBS4 to cilia [153] and RPGRIP1L positively regulates autophagic activity [156]. Thus, it is conceivable that RPGRIP1L ensures the ciliary amount of BBS4 via regulating autophagy. Considering that BBS4 positively regulates proteasomal activity [66], it is possible that RPGRIP1L controls proteasomal activity by affecting the ciliary amount of BBS4. However, BBS4 seems to control overall cellular proteasomal activity [66] while RPGRIP1L governs proteasomal activity exclusively at the ciliary base [63]. Furthermore, it was demonstrated that RPGRIP1L regulates the activity of the cilia-regulated proteasome by interacting with proteasome 26S non-ATPase regulatory subunit 2 (PSMD2) [63], arguing for a BBS4-independent control of proteasomal activity by RPGRIP1L. Another protein whose abundance at the TZ is governed by RPGRIP1L represents nephrocystin 4 (NPHP4), which also participates in the regulation of protein degradation. NPHP4 interacts with an E3 ligase named Jade-1, which targets β-catenin, the main mediator of the canonical WNT pathway, to the proteasome. By this interaction, NPHP4 stabilises Jade-1 and enhances its capability to promote the proteasomal degradation of β-catenin in a human embryonic kidney cell line (HEK293 cells) [159]. A link between NPHP4 and autophagy has not been shown. The E3 ligases c-CBL and CBL-b also interact with a ciliary protein, the intraflagellar transport protein 20 (IFT20). It stabilises c-CBL and CBL-b by inhibiting their autoubiquitination and proteasomal degradation. In this way, IFT20 supports the ubiquitination and internalization of PDGFRα thereby preventing aberrant PDGFRα signalling in immortalised MEFs (NIH3T3 cells) [160]. In the context of autophagy, analyses in MEFs demonstrated that IFT20 is an autophagic substrate and a positive regulator of autophagy suggesting “a novel mechanism for self-containment of the autophagic process” [155]. 

Apart from HH signalling, another signal transduction cascade links cilia to the UPS and autophagy. The mTOR signalling pathway is mediated by primary cilia in a human kidney proximal tubular epithelial cell line (HK2 cells) and in MEFs [154,156] and among many other processes is deeply involved in the regulation of both the UPS and autophagy [161]. It is known that mTOR signalling inhibits autophagy [130,131,132,133,161]. In addition, mTOR signalling negatively regulates proteasomal activity in MEFs and in HEK293 cells [40]. Regarding proteasomal activity, the opposite is also observed, namely that mTOR signalling positively governs proteasomal activity as was shown in MEFs and in HEK293 cells [162], leading to an unresolved situation [163,164].

## 6. Conclusions

The primary cilium plays a vital role in the crosstalk between the UPS and autophagy, but it is impossible to define a blanket function of the primary cilium in this crosstalk since it is a signalling hub and houses numerous proteins. As we have outlined in this article, various ciliary proteins and cilia-mediated signalling pathways have different effects on the two degradation systems. It is the beginning of an enormous puzzle where the bulk of the pieces are missing and future studies have to be undertaken to yield a clear picture.

## Figures and Tables

**Figure 1 cells-08-00241-f001:**
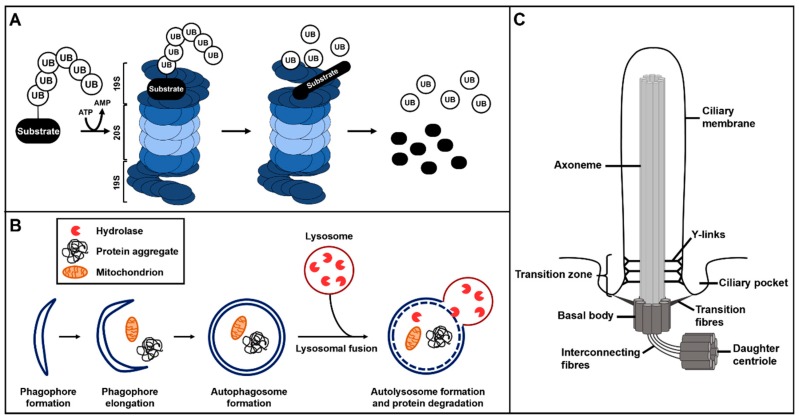
Overview of ubiquitin–proteasome system (UPS) protein degradation, autophagy and primary cilium structure. (**A**) The ubiquitin–proteasome system. The ubiquitinated substrate is recognised by the 19S regulatory subunit of the 26S proteasome and gets degraded or proteolytically processed by the 20S subunit of the 26S proteasome. (**B**) Autophagy starts with the formation of phagophores which subsequently elongate to finally develop into autophagosomes. During autophagosome formation, target proteins and structures become enclosed in the autophagosomes. These autophagosomes fuse with lysosomes and the hydrolases of the lysosomes degrade the target proteins and structures. (**C**) Primary cilia consist of a microtubule scaffold called axoneme. The axoneme is surrounded by the ciliary membrane and grows out of the basal body. The basal body is a modified mother centriole that is connected to the daughter centriole by interconnecting fibres. The basal body is attached to the ciliary membrane in the region of the ciliary pocket via transition fibres. The transition zone, with its Y-links, is located at the proximal part of the axoneme.

**Figure 2 cells-08-00241-f002:**
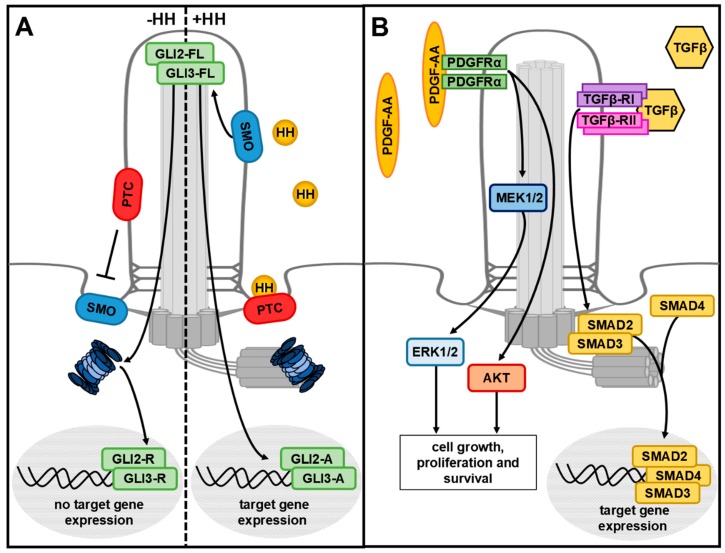
Cilia-mediated HH, PDGFRα and TGFβ signalling. (**A**) Without the HH ligand, the receptor PTC inhibits ciliary entry of SMO. GLI2-FL and GLI3-FL are proteolytically processed into the repressor forms GLI2-R and GLI3-R by the ciliary proteasome. They translocate into the nucleus and inhibit HH target gene expression. In the presence of HH, HH binds to its receptor PTC and the HH/PTC complex leaves the cilium. SMO enters the cilium and GLI2-FL and GLI3-FL become activated and, in turn, initiate HH target gene expression. (**B**) PDGF-AA binds to the ciliary receptor PDGFRα and activates AKT signalling or the MEK1/2–ERK1/2 signalling cascade. TGFβ binds to a heterotetrameric receptor composed of TGFβ-RI and TGFβ-RII in the ciliary membrane. At the base of cilia, the signal is transduced via different SMAD proteins.

**Figure 3 cells-08-00241-f003:**
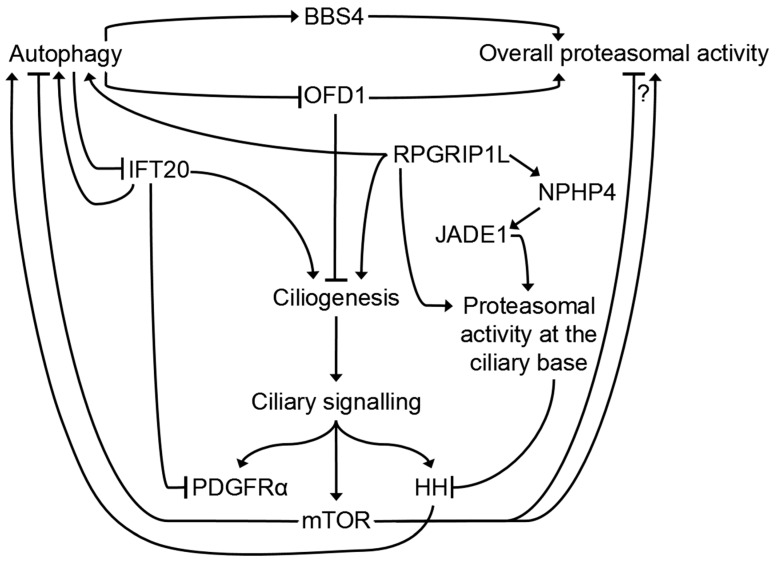
Complex network of the cilia-related crosstalk between the UPS and autophagy. Autophagy is able to regulate overall proteasomal activity in a positive or negative manner via BBS4 or OFD1. In addition, autophagy regulates ciliogenesis via OFD1 and/or IFT20 and thereby affects ciliary signalling. In turn, several cilia-mediated signalling cascades, like HH and mTOR signalling, modulate autophagy. Moreover, mTOR signalling regulates the overall proteasomal activity either positively or negatively (which is a matter of fierce debate). Additional cilia-associated proteins, like IFT20 or RPGRIP1L, regulate ciliogenesis as well as autophagy, whereby RPGRIP1L is also able to regulate the activity of the ciliary proteasome.

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
