# Peer review of "The Role of Primary Cilia in the Crosstalk between the Ubiquitin–Proteasome System and Autophagy"

_cells, 2019, doi:10.3390/cells8030241_

Round 1

Reviewer 1 Report

In this review, the authors focused on the role of primary cilia in the cross-talk between the ubiquitin-proteasome system (UPS) and autophagy-lysosomal pathway (ALP). It is a very good angle of approaching the role of this cellular sub-structure, as there is not much study primary cilia in coordinating two major degradation pathways. The idea is novel but the review is no more than reporting other studies. The review on UPS and autophagy is too general. The figures are not providing any analyzed information from published studies. There are some misleading information about the UPS system. I have to say that this review must be largely rewritten to meet the basic requisite of a scientific review. I don’t recommend it for publication at this stage.

Here are some of my concerns, which may help if authors seek for improving the manuscript:

1.     Line 3, “proteasome-ubiquitin system” in the title: It is commonly used as “ubiquitin-proteasome system (UPS)”, just as the way presented in other places of this manuscripts. It should be written in a right and consistent way.

2.     Line 20-21:  “…as a consequence of improper folded or misfolded proteins.” I believe “misfolded” means “improper folded”.

3.     Line 41-43: I don’t think any targeted proteins need to be phosphorylated before ubiquitination.

4.     Line 46: There are different types E3s. RING domain E3s don’t pass ubiquitin onto E3s.

5.     Line47: Here lacks the formation of polyubiquitin chain.

6.     Line 50: The proteasome recognizes ubiquitin chain, not targeted proteins.

7.     Line 124-174: When reviewing published studies, it is critical to clarify the model systems, tissues, cell lines in which the studies were carried out.

8.     Figure 1, 2, and 3: The three figures are simple re-production of published models from other studies. They lacks novelty and references. It would be a better idea if the authors could draw a model including the ciliary proteins, e.g. BBS4, OFD4, IFT20… and UPS/autophagy, which summarize the content of Line 118-174.

9.     References: Duplicated numbers.

Author Response

Dear Referee,

We would like to thank you for your time and your helpful comments on our manuscript entitled “The role of primary cilia in the cross-talk between the ubiquitin-proteasome system and autophagy” (Manuscript ID: cells-426685). We have taken these comments as a guide for the correction and quality improvement of our manuscript. Every point you made was carefully considered and revised. Your questions/comments are written in italic letters, our answers not.

The following points were made:

In this review, the authors focused on the role of primary cilia in the cross-talk between the ubiquitin-proteasome system (UPS) and autophagy-lysosomal pathway (ALP). It is a very good angle of approaching the role of this cellular sub-structure, as there is not much study primary cilia in coordinating two major degradation pathways. The idea is novel but the review is no more than reporting other studies. The review on UPS and autophagy is too general. The figures are not providing any analyzed information from published studies. There are some misleading information about the UPS system. I have to say that this review must be largely rewritten to meet the basic requisite of a scientific review. I don’t recommend it for publication at this stage.

Here are some of my concerns, which may help if authors seek for improving the manuscript:

1. Line 3, “proteasome-ubiquitin system” in the title: It is commonly used as “ubiquitin-proteasome system (UPS)”, just as the way presented in other places of this manuscripts. It should be written in a right and consistent way.

Answer: We thank you for pointing this out and apologise for the mistake. The title was revised (line 3).

2. Line 20-21: “…as a consequence of improper folded or misfolded proteins.” I believe “misfolded” means “improper folded”.

Answer: We revised the text and deleted “improper folded” (lines 21 and 22).

3. Line 41-43: I don’t think any targeted proteins need to be phosphorylated before ubiquitination.

Answer: We deleted the phosphorylation part since the degradation of most substrates by the UPS does not require phosphorylation (exception: substrates that are recognised through the carboxyl-terminus of F-box proteins and as a result ubiquitin is transferred to the substrates from E2) (lines 49 and 50).

4. Line 46: There are different types E3s. RING domain E3s don’t pass ubiquitin onto E3s.

Answer: Thank you for pointing this out. We modified the text accordingly thereby subdividing RING domain E3 ligases and HECT domain E3 ligases (lines 53-65).

5. Line 47: Here lacks the formation of polyubiquitin chain.

Answer: The process of ubiquitination is integrated into the revised version of the manuscript (lines 59-65).

6. Line 50: The proteasome recognizes ubiquitin chain, not targeted proteins.

Answer: We revised the text (lines 68-70).

7. Line 124-174: When reviewing published studies, it is critical to clarify the model systems, tissues, cell lines in which the studies were carried out.

Answer: The manuscript was modified accordingly, now providing the requested information (lines 240-297).

8. Figure 1, 2, and 3: The three figures are simple re-production of published models from other studies. They lacks novelty and references. It would be a better idea if the authors could draw a model including the ciliary proteins, e.g. BBS4, OFD4, IFT20… and UPS/autophagy, which summarize the content of Line 118-174.

Answer: We merged the first three images in one Figure (Figure 1) in order to provide a simple introduction for the reader and, additionally, we created a new figure (Figure 3) which summarises the content of this section.

9. References: Duplicated numbers.

Answer: The problem is fixed (lines 319-734).

Reviewer 2 Report

This review article should not be published unless extensively revised. The authors are clearly not knowledgeable about the ubiquitin-proteasome pathway and make a number of incorrect statements about it. The statements on protein breakdown are based upon older reviewers and a number of a recent relevant findings are not mentioned. The description of autophagy seems also dated, and statements are imprecise while the second part discussing cilia is much better, the details are often not clearly described, and the observations not evaluated sufficiently critically. In fact, the assumption of “cross talk” is actually not defined carefully and discussed critically.

Major Problems:

1.      Line 2 and onward: The idea that the UPS and autophagy are partially redundant is misleading. Their primary functions, substrates, and subcellular localization are totally distinct. There are hundreds of ubiquitin ligases with distinct functions. So, generalizations about the UPS are dangerous and misleading. In fact, use of proteasome inhibitors causes compensatory increases in proteasome expression and only much later upregulation of autophagy, which may be linked to cell death (see for example Sha et al. J Cell Biol. 2018; 217(5). Doi: 10.1083/jcb.201708168. Also, there is often coordinated slow regulation of these two processes by FoxO transcription factor (See Zhao et al. Cell Metabolism. 2007; 6: 472-483.) and rapid regulation through by mTOR and mutant supply (see Zhao et al. PNAS. 2015; 112(52):15790-7. Doi: 10.1073/pnas1522332112).

2.      Is there any evidence that blocking lysosomes causes an increase in degradation by UPS? In fact, many studies show linear rates of the UPS or lysosomal degradation when the other process is blocked for many hours. So, they seem independent overall. The more recent paper of Rubinsztein and colleagues should be cited on “cross talk”.

Specific Points:

1.      Line 26: The focus on signaling seems misleading. The essential roles of the UPS are in cell cycle regulation and controlling transcription and protein quality control.

2.      Line 40: “Peptide bonds are dissociated from the target protein”. What is meant here? The statement is clearly false as written.

3.      Line 39: Proteolytic processing by the UPS is definitely not a general role of the UPS. There are thousands of proteins degraded processively to small peptides by the proteasome. To my knowledge, limited processing of G11 and NFkB are exceptions and the mechanisms are unclear.

4.      Line 42: No. Degradation of most substrates by the UPS does not require phosphorylation. Only a small fraction, which involve F-box E3s.

5.      Line 46: No, the E2s only transfer Ub to E3s of the HECT family. This is not true for most of the 600 E3s in humans, which do not bind the Ub.

6.      Line 66: This description of autophagy is superficial and out of date. The roles of different ATG genes are not mentioned. Only a fraction of autophagy substrates are ubiquitinated. This is selective autophagy. It is wrong to say Lysine residues do not promote ubiquitination.

7.      What exactly is meant by Cross Talk? A clear definition is given and the various papers (73-85) use the terms differently.

8.      Line 13: How can autophagy recruit a protein to cilia?

9.      Page 138-145: Are these effects on the proteasomes catalytic activities or on proteasome amounts?

10.  Page 140-150: Unclear. Explain how does this cilia-regulated proteasome differ from other cell proteasomes?

11.  Is RPGRIP1L controlling degradation of proteins though effects on proteasome activity or on the ubiquitination step? What happens in cells other than MDCK?

12.  Page 163-165: Doesn’t CBL-target proteins to the endosomal-lysosomal pathway not proteasomes?

13.  Page 168-171: This is not the primary action of mTOR, which has many substrates throughout the cell. The extensive work of Sabatini and colleagues indicates that mTOR associates with lysosomes and also controls autophagy through ULK-1.

14.  I could not find Reference 43. Could it be an error?

Author Response

Dear Referee,

We appreciate your comments on our manuscript entitled “The role of primary cilia in the cross-talk between the ubiquitin-proteasome system and autophagy” (Manuscript ID: cells-426685) which gave us invaluable suggestions for the revision of our manuscript. As detailed below we have addressed your specific comments in a point-by-point manner. Your questions/comments are written in italic letters, our answers not.
The following points were made:

This review article should not be published unless extensively revised. The authors are clearly not knowledgeable about the ubiquitin-proteasome pathway and make a number of incorrect statements about it. The statements on protein breakdown are based upon older reviewers and a number of a recent relevant findings are not mentioned. The description of autophagy seems also dated, and statements are imprecise while the second part discussing cilia is much better, the details are often not clearly described, and the observations not evaluated sufficiently critically. In fact, the assumption of “cross talk” is actually not defined carefully and discussed critically.

Major Problems:

1. Line 2 and onward: The idea that the UPS and autophagy are partially redundant is misleading. Their primary functions, substrates, and subcellular localization are totally distinct. There are hundreds of ubiquitin ligases with distinct functions. So, generalizations about the UPS are dangerous and misleading. In fact, use of proteasome inhibitors causes compensatory increases in proteasome expression and only much later upregulation of autophagy, which may be linked to cell death (see for example Sha et al. J Cell Biol. 2018; 217(5). Doi: 10.1083/jcb.201708168. Also, there is often coordinated slow regulation of these two processes by FoxO transcription factor (See Zhao et al. Cell Metabolism. 2007; 6: 472-483.) and rapid regulation through by mTOR and mutant supply (see Zhao et al. PNAS. 2015; 112(52):15790-7. Doi: 10.1073/pnas1522332112).

2. Is there any evidence that blocking lysosomes causes an increase in degradation by UPS? In fact, many studies show linear rates of the UPS or lysosomal degradation when the other process is blocked for many hours. So, they seem independent overall. The more recent paper of Rubinsztein and colleagues should be cited on “cross talk”.

Answer: Since the two major problems are closely related to each other, we provide a combined answer to both points. We are very surprised that your comments seem to challenge the existence of a cross talk between the UPS and autophagy, in particular because the information text of the special issue, our manuscript is written for, is the following: “Proteins belong to the most skilled but unstable components of the cell and their correct folding, ensuring the correct three-dimensional structure, is the basis of their function. Misfolding of proteins during translation, or as a consequence of internal or external challenges, calls for refolding mechanisms. Finally, if refolding fails, the rapid degradation of the target protein is mandatory. Therefore, folding, refolding, and degradation are the pillars of protein homeostasis (proteostasis) that needs to be tightly controlled to maintain proper cellular functions. A great effort has been made to understand the regulators of proteostasis, which has resulted in the definition of a fine-tuned network of factors, including molecular chaperones and the two main protein degradation routes, the ubiquitin proteasome system and the autophagy-lysosomal pathways. Recently, it became evident that both degradation
pathways are not working independently of each other, but that there is cross-talk between them. While proteasomal degradation is rather well-defined, data on autophagic degradation of proteins and also intracellular organelles are currently exploding, precisely because there are many links of changes in autophagy to diseases, including neurodegeneration. The maintenance of cellular proteostasis and, in particular, the adequate degradation and removal of dysfunctional proteins is of distinct importance for post-mitotic cells such as neurons.” (https://www.mdpi.com/journal/cells/special_issues/proteostasis_autophagy). Our surprise is even increased when one considers that several recent review articles define the kind of cross talk as follows: “Until recently, the UPS and autophagy were considered two parallel protein degradation machineries with no point of intersection (Korolchuk et al., 2009). This idea was fostered partly because autophagy and the UPS have separate molecular machinery and substrate preferences (Korolchuk et al., 2010). Autophagy is a vesicular trafficking pathway that specializes in the delivery of long-lived proteins and damaged organelles to the lysosome (Klionsky et al., 2008). The degradation of soluble, short-lived regulatory proteins by the UPS, on the other hand, occurs in the cytosol (Streich and Lima, 2014). According to the classical definition, the UPS is a selective degradation process for cellular proteins that require temporal control, such as regulatory and cell cycle-related proteins (Ciechanover, 2005). Autophagy, in contrast, is viewed as a cellular response that serves to scavenge nutrients when cells are subjected to starvation (Russell et al., 2014). However, from the cellular point of view, it would make sense that the two major protein-degradation machineries, with their implications in cellular homeostasis, communicate with each other. In line with this theory, studies conducted in the last decade have irrefutably confirmed this paradigm by unraveling the interplay between autophagy and the UPS at the molecular and functional levels (Dikic, 2017). Possibly the strongest link between the UPS and autophagy comes from the observation that several molecules are shared as either regulators or substrates of both these pathways (Lilienbaum, 2013).” (Nam et al., 2017); “Since the inception of their discoveries, the UPS and autophagy were thought to be independent of each other in components, action mechanisms, and substrate selectivity. Recent studies suggest that cells operate a single proteolytic network comprising of the UPS and autophagy that share notable similarity in many aspects and functionally cooperate with each other to maintain proteostasis. … Central to both the UPS and autophagy is ubiquitination, i.e., the conjugation of the 76-amino acid protein ubiquitin to the lysine (Lys) residues of other proteins (Ciechanover, 2015; Ciechanover and Kwon, 2015; Ciechanover and Kwon, 2017).” (Ji and Kwon, 2017); “Autophagy and the ubiquitin–proteasome system (UPS) are the two major intracellular quality control and recycling mechanisms that are responsible for cellular homeostasis in eukaryotes. Ubiquitylation is utilized as a degradation signal by both systems, yet, different mechanisms are in play. … Initial observations about functional connections between the UPS and autophagy systems revealed that inhibition of one led to a compensatory upregulation of the other system. In order to maintain homeostasis, cellular materials that accumulate following inhibition of one degradative system needs to be cleared, at least in part, by the other system.” (Kocaturk and Gozuacik, 2018). Consequently, there are multiple studies which confirm the existence of a cross talk between the UPS and autophagy and which state that these two protein degradation pathways at least partially share the same substrates.

Based on this, we tried to follow your argumentation regarding the kind of this cross talk by dissecting several research papers. As your examples point out, the reduction of proteasomal activity leads to an increase in autophagic activity. Thus, all these data argue for the existence of a proteasome-to-autophagy direction of regulation. And there are many more examples for this kind of direction (Demishtein et al., 2017; Fan et al., 2018; Ge et al., 2009; Jiang et al., 2015; Kyrychenko et al., 2014; Selimovic et al., 2013; Sun et al., 2016; Tang et al., 2014; Wu et al., 2008; Xu et al., 2012; Zhu et al., 2010). But there is also evidence from two studies that a decreased autophagic activity can be compensated by an increase of proteasomal activity (Tannous et al., 2008; Wang et al., 2013). To take your criticism into account, we revised the manuscript and clarified that the proteasome-to-autophagy direction of regulation is far better documented than the autophagy-to-proteasome direction (lines 34-38).

Specific Points:
1. Line 26: The focus on signaling seems misleading. The essential roles of the UPS are in cell cycle regulation and controlling transcription and protein quality control.

Answer: Thank you for mentioning. We revised the manuscript accordingly (lines 26 and 27).

2. Line 40: “Peptide bonds are dissociated from the target protein”. What is meant here? The statement is clearly false as written.

Answer: Thank you for pointing this out. We revised the sentence accordingly (lines 47 and 48).

3. Line 39: Proteolytic processing by the UPS is definitely not a general role of the UPS. There are thousands of proteins degraded processively to small peptides by the proteasome. To my knowledge, limited processing of G11 and NFkB are exceptions and the mechanisms are unclear.

Answer: Proteolytic processing is an important process for the transduction of signals in the context of signalling pathways that are essential for development and homeostasis (Gerhardt et al., 2016). Since these pathways play an important role in the further course of the manuscript, the reader urgently needs to know about the ability of the UPS to realise proteolytic processing. Due to this point, we introduced proteolytic processing at this place within the text.

4. Line 42: No. Degradation of most substrates by the UPS does not require phosphorylation. Only a small fraction, which involve F-box E3s.

Answer: We are grateful for this information and deleted the phosphorylation section (lines 49 and 50).

5. Line 46: No, the E2s only transfer Ub to E3s of the HECT family. This is not true for most of the 600 E3s in humans, which do not bind the Ub.

Answer: We agree and revised the text accordingly (lines 53-59).

6. Line 66: This description of autophagy is superficial and out of date. The roles of different ATG genes are not mentioned. Only a fraction of autophagy substrates are ubiquitinated. This is selective autophagy. It is wrong to say Lysine residues do not promote ubiquitination.

Answer: In the new version of our manuscript, we described autophagy in more detail by including the functions of the different ATG proteins (lines 101-121). Moreover, we introduced selective autophagy (lines 122 and 123). In the context of lysine residues, we modified the text, now providing the statement: “Ubiquitin has different lysine residues which participate in the generation of polyubiquitin chains” (lines 123 and 124).

7. What exactly is meant by Cross Talk? A clear definition is given and the various papers (73-85) use the terms differently.

Answer: To avoid any information that is out of date, we only cite the most recent review articles in the new version of the manuscript. These reviews are in line with our definition of the cross-talk between the UPS and autophagy: “The key finding was that the UPS and autophagy act at least partially redundant, a result that is based on the facts that both degradation systems partially share the same substrates and that if one degradation system is downregulated, the other gets upregulated. It was shown that numerous proteins participate in this cross-talk.” (lines 230-232).

8. Line 13: How can autophagy recruit a protein to cilia?

Answer: Tang and colleagues analysed the relationship between primary cilia and autophagy. In this context, they showed that “BBS4 recruitment to primary cilia is defective in Atg5−/− MEFs” (Tang et al., 2013). To avoid any misunderstanding, we added “most likely via an indirect mechanism” (lines 243 and 244).

9. Page 138-145: Are these effects on the proteasomes catalytic activities or on proteasome amounts?

Answer: Previously, we showed that the amount of several proteasomal components is increased in the absence of RPGRIP1L (Gerhardt et al., 2015). However, RPGRIP1L deficiency results in a reduced proteasomal activity at the ciliary base. Thus, RPGRIP1L regulates the activity of the cilia-regulated proteasome most likely by governing catalytic activities. For this reason, we always wrote in the manuscript that RPGRIP1L regulates proteasomal activity. To make this clearer, we added the word “catalytic” (lines 252-261).

10. Page 140-150: Unclear. Explain how does this cilia-regulated proteasome differ from other cell proteasomes?

Answer: By using the Proteasome Sensor Vector, we were able to show that RPGRIP1L deficiency affects the activity of the cilia-regulated proteasome but not of all other proteasomes within the cell. We modified the text in order to explain how the difference between the cilia-regulated proteasome and other proteasomes was found (lines 250-255).

11. Is RPGRIP1L controlling degradation of proteins though effects on proteasome activity or on the ubiquitination step? What happens in cells other than MDCK?

Answer: As written in the manuscript (“RPGRIP1L governs proteasomal activity exclusively at the ciliary base (Gerhardt et al., 2015). Furthermore, it was demonstrated that RPGRIP1L regulates the activity of the cilia-regulated proteasome by interacting with Proteasome 26S Subunit, Non-ATPase 2 (PSMD2) (Gerhardt et al., 2015)”) (lines 274 and 275), RPGRIP1L interacts with PSMD2 arguing for proteasomal activity control rather than for regulating ubiquitination. To avoid misunderstandings, we included the respective system in which the analyses were performed.

12. Page 163-165: Doesn’t CBL-target proteins to the endosomal-lysosomal pathway not proteasomes?

Answer: Based on the findings of Schmid et al. (Schmid et al., 2018), we wrote that IFT20 “stabilises c-CBL and CBL-b by inhibiting their autoubiquitination and proteasomal degradation. In this way, IFT20 supports the ubiquitination and internalization of PDGFRα thereby preventing aberrant PDGFRα signalling in immortalised MEFs (NIH3T3 cells)” (lines 283-286). Thus, c-CBL and CBL-b are protected from proteasomal degradation by their interaction with IFT20 and they are involved in the ubiquitination and internalization of PDGFRα.

13. Page 168-171: This is not the primary action of mTOR, which has many substrates throughout the cell. The extensive work of Sabatini and colleagues indicates that mTOR associates with lysosomes and also controls autophagy through ULK-1.

Answer: We revised the text accordingly (lines 115-121 and 293).

14. I could not find Reference 43. Could it be an error?

Answer: In the revised version of the manuscript, all references are included.

References:
Ciechanover, A. 2005. Proteolysis: from the lysosome to ubiquitin and the proteasome. Nat. Rev. Mol. Cell Biol. 6:79-87.
Ciechanover, A. 2015. The unravelling of the ubiquitin system. Nat. Rev. Mol. Cell Biol. 16:322-324.
Ciechanover, A., and Y. Kwon. 2015. Degradation of misfolded proteins in neurodegenerative diseases: therapeutic targets and strategies. Exp. Mol. Med. 47:e147.
Ciechanover, A., and Y. Kwon. 2017. Protein Quality Control by Molecular Chaperones in Neurodegeneration. Front. Neurosci. 11:185.
Demishtein, A., M. Fraiberg, D. Berko, B. Tirosh, Z. Elazar, and A. Navon. 2017. SQSTM1/p62-mediated autophagy compensates for loss of proteasome polyubiquitin recruiting capacity. Autophagy. 13:1697-1708.
Dikic, I. 2017. Proteasomal and Autophagic Degradation Systems. Annu. Rev. Biochem. 86:193-224.
Fan, T., Z. Huang, W. Wang, B. Zhang, Y. Xu, Z. Mao, L. Chen, H. Hu, and Q. Geng. 2018. Proteasome inhibition promotes autophagy and protects from endoplasmic reticulum stress in rat alveolar macrophages exposed to hypoxia-reoxygenation injury. J. Cell. Physiol. 233:6748-6758.
Ge, P., J. Zhang, X. Wang, F. Meng, W. Li, Y. Luan, F. Ling, and Y. Luo. 2009. Inhibition of autophagy induced by proteasome inhibition increases cell death in human SHG-44 glioma cells. Acta. Pharmacol. Sin. 30:1046-1052.
Gerhardt, C., T. Leu, J. Lier, and U. Rüther. 2016. The cilia-regulated proteasome and its role in the development of ciliopathies and cancer. Cilia. 5:14.
Gerhardt, C., J. Lier, S. Burmühl, A. Struchtrup, K. Deutschmann, M. Vetter, T. Leu, S. Reeg, T. Grune, and U. Rüther. 2015. The transition zone protein Rpgrip1l regulates proteasomal activity at the primary cilium. J. Cell Biol. 210:115-133.
Ji, C., and Y. Kwon. 2017. Crosstalk and Interplay between the Ubiquitin-Proteasome System and Autophagy. Mol. Cells. 40:441-449.
Jiang, S., D. Park, Y. Gao, S. Ravi, V. Darley-Usmar, E. Abraham, and J. Zmijewski. 2015. Participation of proteasome-ubiquitin protein degradation in autophagy and the activation of AMP-activated protein kinase. Cell. Signal. 27:1186-1197.
Klionsky, D., H. Abeliovich, P. Agostinis, D. Agrawal, G. Aliev, D. Askew, M. Baba, E. Baehrecke, B. Bahr, A. Ballabio, B. Bamber, D. Bassham, E. Bergamini, X. Bi, M. Biard-Piechaczyk, J. Blum, D. Bredesen, J. Brodsky, J. Brumell, U. Brunk, W. Bursch, N. Camougrand, E. Cebollero, F. Cecconi, Y. Chen, L. Chin, A. Choi, C. Chu, J. Chung, P. Clarke, R. Clark, S. Clarke, C. Clavé, J. Cleveland, P. Codogno, M. Colombo, A. Coto-Montes, J. Cregg, A. Cuervo, J. Debnath, F. Demarchi, P. Dennis, P. Dennis, V. Deretic, R. Devenish, F. Di Sano, J. Dice, M. Difiglia, S. Dinesh-Kumar, C. Distelhorst, M. Djavaheri-Mergny, F. Dorsey, W. Dröge, M. Dron, W.J. Dunn, M. Duszenko, N. Eissa, Z. Elazar, A. Esclatine, E. Eskelinen, L. Fésüs, K. Finley, J. Fuentes, J. Fueyo, K. Fujisaki, B. Galliot, F. Gao, D. Gewirtz, S. Gibson, A. Gohla, A. Goldberg, R. Gonzalez, C. González-Estévez, S. Gorski, R. Gottlieb, D. Häussinger, Y. He, K. Heidenreich, J. Hill, M. Høyer-Hansen, X. Hu, W. Huang, A. Iwasaki, M. Jäättelä, W. Jackson, X. Jiang, S. Jin, T. Johansen, J. Jung, M. Kadowaki, C. Kang, A. Kelekar, D. Kessel, J. Kiel, H. Kim, A. Kimchi, T. Kinsella, K. Kiselyov, K. Kitamoto, E. Knecht, et al. 2008. Guidelines for the use and interpretation of assays for monitoring autophagy in higher eukaryotes. Autophagy. 4:151-175.
Kocaturk, N., and D. Gozuacik. 2018. Crosstalk Between Mammalian Autophagy and the Ubiquitin-Proteasome System. Front. Cell Dev. Biol. 6:128.
Korolchuk, V., F. Menzies, and D. Rubinsztein. 2009. A novel link between autophagy and the ubiquitin-proteasome system. Autophagy. 5:862-863.
Korolchuk, V., F. Menzies, and D. Rubinsztein. 2010. Mechanisms of cross-talk between the ubiquitin-proteasome and autophagy-lysosome systems. FEBS Lett. 584:1393-1398.
Kyrychenko, V., V. Nagibin, L. Tumanovska, D. Pashevin, V. Gurianova, A. Moibenko, V. Dosenko, and D. Klionsky. 2014. Knockdown of PSMB7 induces autophagy in cardiomyocyte cultures: possible role in endoplasmic reticulum stress. Pathobiology. 81:8-14.
Lilienbaum, A. 2013. Relationship between the proteasomal system and autophagy. Int. J. Biochem. Mol. Biol. 4:1-26.
Nam, T., J. Han, S. Devkota, and H. Lee. 2017. Emerging Paradigm of Crosstalk between Autophagy and the Ubiquitin-Proteasome System. Mol. Cells. 40:897-905.
Russell, R., H. Yuan, and K. Guan. 2014. Autophagy regulation by nutrient signaling. Cell Res. 24:42-57.
Schmid, F., K. Schou, M. Vilhelm, M. Holm, L. Breslin, P. Farinelli, L. Larsen, J. Andersen, L. Pedersen, and S. Christensen. 2018. IFT20 modulates ciliary PDGFRα signaling by regulating the stability of Cbl E3 ubiquitin ligases. J. Cell Biol. 217:151-161.
Selimovic, D., B. Porzig, A. El-Khattouti, H. Badura, M. Ahmad, F. Ghanjati, S. Santourlidis, Y. Haikel, and M. Hassan. 2013. Bortezomib/proteasome inhibitor triggers both apoptosis and autophagy-dependent pathways in melanoma cells. Cell. Signal. 25:308-318.
Streich, F.J., and C. Lima. 2014. Structural and functional insights to ubiquitin-like protein conjugation. Annu. Rev. Biophys. 43:357-379.
Sun, A., C. Li, R. Chen, Y. Huang, Q. Chen, X. Cui, H. Liu, J. Thrasher, and B. Li. 2016. GSK-3β controls autophagy by modulating LKB1-AMPK pathway in prostate cancer cells. Prostate. 76:172-183.
Tang, B., J. Cai, L. Sun, Y. Li, J. Qu, B. Snider, and S. Wu. 2014. Proteasome inhibitors activate autophagy involving inhibition of PI3K-Akt-mTOR pathway as an anti-oxidation defense in human RPE cells. PLoS One. 9:e103364.
Tang, Z., M. Lin, T. Stowe, S. Chen, M. Zhu, T. Stearns, B. Franco, and Q. Zhong. 2013. Autophagy promotes primary ciliogenesis by removing OFD1 from centriolar satellites. Nature. 502:254-257.
Tannous, P., H. Zhu, A. Nemchenko, J. Berry, J. Johnstone, J. Shelton, F.J. Miller, B. Rothermel, and J. Hill. 2008. Intracellular protein aggregation is a proximal trigger of cardiomyocyte autophagy. Circulation. 117:3070-3078.
Wang, X., J. Yu, S. Wong, A. Cheng, F. Chan, S. Ng, C. Cho, J. Sung, and W. Wu. 2013. A novel crosstalk between two major protein degradation systems. Autophagy. 9:1500-1508.
Wu, W., Y. Wu, L. Yu, Z. Li, J. Sung, and C. Cho. 2008. Induction of autophagy by proteasome inhibitor is associated with proliferative arrest in colon cancer cells. Biochem. Biophys. Res. Commun. 374:258-263.
Xu, J., S. Wang, B. Viollet, and M. Zou. 2012. Regulation of the proteasome by AMPK in endothelial cells: the role of O-GlcNAc transferase (OGT). PLoS One. 7:e36717.
Zhu, K., K.J. Dunner, and D. McConkey. 2010. Proteasome inhibitors activate autophagy as a cytoprotective response in human prostate cancer cells. Oncogene. 29:451-462.

Reviewer 3 Report

In the present review, the authors described the role of primary cilia in the regulation of equilibrium between the UPS and autophagy. 

The review is well constructed introducing the description of UPS, autophagy and primary cilia, and afterwards the cross-talk among the degradative systems, with particular regard to primary cilia.

The review sounds interesting especially because it describes important pathways in primary cilia, fundamental organelles of cell sensory process, also involved in a variety of human disorders.  

I suggest a revision of the present review.

In particular, there are some points that need to be improved or revised:

-      Line 42, the Authors assert that degradation by UPS starts with phosphorylation.

It is known an interplay between phosphorylation and ubiquitination, but phosphorylation is not mandatory for ubiquitination. I suggest to revise this part or to provide the proper references to support this sentence.

-      Paragraph 3, I suggest to extend this paragraph adding further information about the functions of primary cilia. Moreover, I suggest to add a figure representing the mechanisms of signaling in primary cilia described in the paragraph.  

-      I suggest to add a further paragraph to describe the role of UPS and autophagy in ciliopathies.

Author Response

Dear Referee,

We would like to thank you for taking the time to carefully read and comment on our manuscript, which we have now revised in accordance with your valuable comments (Manuscript ID: cells-426685). In the following we have included our response to each point you raised. Your questions/comments are written in italic letters, our answers not.

The following points were made:

In the present review, the authors described the role of primary cilia in the regulation of equilibrium between the UPS and autophagy.
The review is well constructed introducing the description of UPS, autophagy and primary cilia, and afterwards the cross-talk among the degradative systems, with particular regard to primary cilia.
The review sounds interesting especially because it describes important pathways in primary cilia, fundamental organelles of cell sensory process, also involved in a variety of human disorders.

I suggest a revision of the present review.
In particular, there are some points that need to be improved or revised:

- Line 42, the Authors assert that degradation by UPS starts with phosphorylation.

It is known an interplay between phosphorylation and ubiquitination, but phosphorylation is not mandatory for ubiquitination. I suggest to revise this part or to provide the proper references to support this sentence.

Answer: We deleted the phosphorylation part since the degradation of most substrates by the UPS does not require phosphorylation (exception: substrates that are recognised through the carboxyl-terminus of F-box proteins and as a result ubiquitin is transferred to the substrates from E2) (lines 49 and 50).

- Paragraph 3, I suggest to extend this paragraph adding further information about the functions of primary cilia. Moreover, I suggest to add a figure representing the mechanisms of signaling in primary cilia described in the paragraph.

Answer: We revised the manuscript by describing the role of cilia in additional signalling pathways (PDGFRα and TGFβ signalling) (lines 160-171) and generated a new figure that illustrates the mediation of these signalling pathways by primary cilia.

- I suggest to add a further paragraph to describe the role of UPS and autophagy in ciliopathies.

Answer: A new paragraph addressing a potential role of the UPS and autophagy in the development of ciliopathies was inserted into the revised version of the manuscript (lines 196-224).

Round 2

Reviewer 1 Report

This revised manuscript has been largely rewritten. The misleading information in UPS has been corrected. The new figure 3 is a good illustration of the complicated network. In general, the revised version has been greatly improved. The authors have addressed concerns from my previous comments. The valuable suggestions from other reviewers have also been taken to make it a better scientific review.

Here are some of my concerns, which are minor in nature.

1.     Line 49: confusing word “realised…”

2.     Figure 2: There is legend for Figure 2C, but there is no related figure.

3.     The title of Figure 3: “Complex network of the cross-talk between UPS, autophagy and cilia.” The wording is confusing.

Author Response

Dear Referee,

We would like to thank you once again for taking the time to carefully read and comment on our manuscript The role of primary cilia in the cross-talk between the ubiquitin-proteasome system and autophagy” (Manuscript ID: cells-426685), which we have now revised in accordance with your valuable comments. We have addressed your specific comments in a point-by-point manner.

Your questions/comments are written in italic letters, our answers not.

The following points were made:

This revised manuscript has been largely rewritten. The misleading information in UPS has been corrected. The new figure 3 is a good illustration of the complicated network. In general, the revised version has been greatly improved. The authors have addressed concerns from my previous comments. The valuable suggestions from other reviewers have also been taken to make it a better scientific review.

Here are some of my concerns, which are minor in nature.

1.      Line 49: confusing word “realised…”

We replaced “realised” with “carried out” (line 46).

2.      Figure 2: There is legend for Figure 2C, but there is no related figure.

We apologise for the mistake and deleted C since both PDGFRα and TGFβ signalling are depicted in Figure 2B.

3.      The title of Figure 3: “Complex network of the cross-talk between UPS, autophagy and cilia.” The wording is confusing.

                    We revised the heading of Figure 3. In the revised version of our manuscript, the new                       heading is: “Complex network of the cilia-related cross-talk between the UPS and                             autophagy”.